# Continuous Glucose Monitoring—New Diagnostic Tool in Complex Pathophysiological Disorder of Glucose Metabolism in Children and Adolescents with Obesity

**DOI:** 10.3390/diagnostics14242801

**Published:** 2024-12-13

**Authors:** Marko Simunovic, Marko Kumric, Doris Rusic, Martina Paradzik Simunovic, Josko Bozic

**Affiliations:** 1Department of Pediatrics, University Hospital of Split, Spinciceva 1, 21000 Split, Croatia; 2Department of Pediatrics, University of Split School of Medicine, Soltanska 2, 21000 Split, Croatia; 3Department of Pathophysiology, University of Split School of Medicine, Soltanska 2, 21000 Split, Croatia; 4Laboratory for Cardiometabolic Research, University of Split School of Medicine, Soltanska 2, 21000 Split, Croatia; 5Department of Pharmacy, University of Split School of Medicine, Soltanska 2, 21000 Split, Croatia; 6Department of Ophthalmology, University Hospital of Split, Spinciceva 1, 21000 Split, Croatia

**Keywords:** continuous glucose monitoring, children, adolescents, obesity

## Abstract

Obesity is one of the leading causes of chronic diseases, and its prevalence is still rising in children and adolescent populations. Chronic cardiovascular complications result in metabolic syndrome (MS) and type 2 diabetes mellitus. Key factors in the development of MS are insulin resistance and low-grade inflammation. The disorder of glucose and insulin metabolism has not been fully elucidated so far, and an oral glucose tolerance test (OGTT) has been the only tool used to look into the complex metabolism disorder in children and adolescents with obesity. Continuous glucose monitoring (CGM) has become commercially available for over two decades and is primarily used to manage type 1 diabetes mellitus in pediatric populations. This review aims to present the current knowledge about the use of CGM in children and adolescent populations with obesity. CGM systems have the potential to serve as valuable tools in everyday clinical practices, not only in the better diagnosis of chronic complications associated with obesity, but CGM can also assist in interventions to make better adjustments to nutritional and therapeutic approaches based on real-time glucose monitoring data. Despite these promising benefits, further research is needed to fully understand the role of CGM in metabolic disorders in pediatric populations with obesity, which will additionally strengthen the importance of CGM systems in everyday clinical practices.

## 1. Introduction

The global pandemic of pediatric obesity is one of the leading causes of chronic diseases in children and adolescent populations [1,2,3]. According to the World Obesity Federation, in 2025, there will be more than 250 million children and adolescents with obesity worldwide [1,4]. Early-life obesity is a significant risk factor for the development of chronic cardiovascular complications and type 2 diabetes mellitus (T2DM) in early adulthood [5,6,7,8]. Unfortunately, chronic cardio-metabolic complications are already visible at an early age, which is one of the key factors in the development of metabolic syndrome (MS) [5,9,10]. MS consists of a cascade of cardio-metabolic complications, including abdominal obesity, hypertension, dyslipidemia, and impaired glucose metabolism, that affect all organ systems in children and adolescents with obesity [5,9,11,12]. Insulin resistance (IR) and low-grade inflammation play a major role in the pathophysiology of the onset and progression of MS [2,9,10,11,13]. IR is the first step in the gradual disruption of glucose metabolism [2,7]. The gold standard in the assessment of IR is the euglycemic-hyperinsulinemic clamp, which is an experimental method and has never been used in everyday clinical practice [14,15]. Most authors estimate IR indirectly based on basal glucose and insulin levels, most often using the homeostasis model for the assessment of insulin resistance (HOMA-IR) [2,14,16]. Furthermore, the oral glucose tolerance test (OGTT) has so far been the only tool used to look into the complex metabolism of glucose and insulin over a longer period of time, but only during the visit to the outpatient clinic [14]. Unfortunately, most of the day is not covered by OGTT, and the impact of the complex pathophysiological cascade of glucose metabolism disorders on early complications in children and adolescents with obesity is still not fully elucidated.

Continuous glucose monitoring (CGM) became commercially available more than 20 years ago, and it is considered one of the main milestones in today’s adequate multidisciplinary treatment of type 1 diabetes mellitus (T1DM) [17,18]. According to the newly available International Society for Pediatric and Adolescent Diabetes (ISPAD) guidelines from 2022, every child and adolescent with diabetes should have access to one of the CGM systems from the very beginning of the disease [17,19]. Additionally, recent studies show that more than 75% of young patients with T1DM use CGM systems today in the developed world [19,20]. Today, more and more new CGM technologies are being used in adult patients with T2DM, many of whom are also obese, in order to better optimize therapy and nutrition [21,22].

The aim of this review is to summarize the current knowledge about the use of CGM in the population of children and adolescents with obesity. Second, based on the current published studies, our goal is to further clarify the complex pathophysiological disorder of glucose metabolism in children and adolescents with obesity and its possible effect on chronic complications.

## 2. Obesity and Metabolic Syndrome in Pediatric Population

According to the Word Health Organization, obesity in children older than 5 years is defined on the basis of the body mass index (BMI) and its standard deviation score (SDS); a child or adolescent is considered obese if the BMI SDS is greater than 2, adjusted for sex and age [9,23,24]. Unfortunately, other key organizations, including the United States Centers for Disease Control and Prevention (CDC) and the International Obesity Task Force (IOTF), use different definitions and cut-off points, which can greatly affect the prevalence itself and also the comparability of the data [1]. Furthermore, more and more attention is being given to the recognition of targeted obesity, such as abdominal obesity, which is considered a separate endocrine organ that secretes specific biomolecules that significantly affect the metabolic balance [25].

The prevalence of obesity is gradually reaching its peak in high-income countries, while unfortunately, the obesity pandemic is gradually affecting the rest of the world [1]. In a large study in 33 European countries on 411,000 children aged 7 to 9 years, the total prevalence of obesity was 12%, while the highest prevalence was in the Mediterranean region in Cyprus, Italy, Greece, and Spain [26]. These results are extremely disappointing, considering that the Mediterranean diet is one of the healthiest dietary patterns [26,27,28]. Unfortunately, adherence to the Mediterranean diet is decreasing, and in combination with inadequate sports activity and an increase in time spent in front of the screen, is a key factor in the development of early obesity [26,27,29]. The results of a recent study from the United States are also not promising; in the population up to 2 to 19 years of age, the total prevalence of obesity is 19.7%, and the results are even more disappointing for some ethnic groups [30].

Early-life obesity gradually affects the development of a pro-inflammatory state that affects all organ systems, which further exacerbates IR. Together, these are the basis of MS and, consequently, T2DM [9,31,32,33,34]. The definition of MS in childhood is extremely demanding, and there are different views on which components and thresholds should be included in the definitions [24,35,36]. According to the leading definitions by the International Diabetes Federation (IDF) and Adult Treatment Panel III (ATP-III), which are adapted definitions from adulthood, mandatory component is abominable obesity with 2 or more of the following criteria: impaired glucose tolerance, low HDL, elevated triglycerides, elevated systolic blood pressure or diastolic blood pressure [11,12]. Interestingly, IR was a criterion in the definition of some of the older definitions, including from the WHO, but today’s definitions do not include it as a mandatory component of MS, and the role of IR in the onset and progression of MS is still unclear [12]. According to a recently published review article, the prevalence of MS in the pediatric population with obesity ranges between 0.3 and 26.4%, depending on the definition and the studied population [37]. Additionally, in a large cohort of more than five thousand pediatric subjects, it was demonstrated that the metabolic disorder occurs at an early age, as early as 5 years of life, and a child with obesity has a 2.4-fold risk for adult MS [38]. Furthermore, the occurrence of MS in childhood significantly increases the carotid thickens of the intima-media, which significantly increases the cardiovascular risk in adulthood [37].

## 3. Glucose Metabolism in Children and Adolescents with Obesity

Although the association between obesity and impaired glucose metabolism is undisputed, the mechanisms explaining it are complex and multifaceted [39,40,41]. First, genetic predisposition to obesity seems to be associated with the risk of T2DM independent of BMI, dietary, and lifestyle risk factors [42,43]. In line with this, separate authors showed that IR may develop at a very early age and that it is associated with a higher expression of genes related to obesity and T2DM [44]. Besides genetic predisposition, the expansion of adipose tissue and consequent microenvironment changes represent the principal culprits in the development of obesity-related hyperglycemia [45]. In short, adipose tissue accumulation accentuates cellular stress and pro-inflammatory perturbations, resulting in increased lipolysis, which dampens the activity of glycogen synthase and indulges gluconeogenesis in the liver by diacylglycerol (DAG)-evoked activation of protein kinase C [46]. In skeletal muscle, an increase in DAG leads to a reduction in glucose uptake, further supporting hyperglycemia. Meanwhile, long-term hyperinsulinemia results in pancreatic β-cell loss due to exhaustion, consequently reducing insulin secretion [47]. Finally, despite an overabundance of calories, children with obesity are commonly burdened by malnutrition in vitamins A, D, C, and E, zinc, and other micronutrients. This further contributes to the development of systemic inflammation and IR [27,48,49,50]. Generally, the link between chronic low-grade inflammation and obesity is one of the most extensively investigated mechanisms in diabetes pathophysiology, and details concerning it are described elsewhere [45].

Early reports highlighted the importance of abdominal obesity in T2DM, but whether glucose regulation significantly depends on it is still a matter of debate [51]. Lu et al. recently demonstrated that both overall and central obesity serve as markers for insulin resistance in childhood [52]. However, they found that while overall obesity is positively linked to hyperglycemia among children with mothers diagnosed with gestational diabetes mellitus (GDM), central obesity alone does not exhibit such an association.

Nonetheless, it is worth noting that the current body of research concerning childhood obesity and early-onset diabetes mellitus primarily originates from high-income countries, with fairly limited representation from low- and middle-income nations [53]. Such disparity limits the generalizability of findings related to the association between obesity and early-onset T2DM; studies involving diverse populations are warranted for more comprehensive assessment.

## 4. Use of CGM in the Pediatric Population

The first commercially available sensor for continuous glucose measurement was approved by regulatory authorities in 1999 [18]. Until then, adequate glycemic control was based on glycated hemoglobin (HbA1c) and results from fingerstick capillary glucose measurement [19]. Since then, many new CGM devices have become available, which have improved over time and are divided by the type of CGM depending on glycemic monitoring in real-time or intermittent scan [17,18]. In addition, CGM technology bases measurements on the level of glycemia from the extracellular fluid, and the measurement interval, reliability, and duration of the sensor have been significantly improved [17]. Measuring adequate glycemia via CGM requires an enzyme reaction located at the tip of the sensor filament that is inserted into the subcutaneous tissue, and so far, numerous substances and drugs have been described that can interfere with the sensor (hydroxyurea, acetaminophen, and ascorbic acid) [54]. Today, there are many CGM sensors commercially available, and the duration of the sensor has been significantly extended, lasting 6 to 14 days on average [55,56,57,58]. Moreover, implantable sensors with extended durations are gradually appearing, and today, a sensor lasting 180 days has been approved in the pediatric population [59,60]. It is important to emphasize that the accuracy of the CGM system as an independent device without adjustments is considered if the mean absolute relative differences (MARD) are less than 10% [61,62]. Detailed specifications of sensors approved in the pediatric population are presented in Table 1. Furthermore, a dose of caution is needed when using CGM because some real-life studies demonstrated significantly greater deviations in MARD in patients with T1DM [63]. Finally, the significantly increased availability and price ratio of CGM sensors significantly changed the therapeutic approach in the CGM era; CGM data is used as the main factor for adequate titration of insulin therapy [64,65].

In addition, there is more and more available data on the use of the CGM system in healthy subjects, further strengthening the current knowledge about the importance of understanding the complex metabolism of glucose and the establishment of adequate normative for future studies [72,73,74]. A recent study by Afeef et al. demonstrated a possible alternative with a CGM sensor to fingerstick capillary glucose measurement during the OGTT in healthy adolescents, which further establishes the strength and quality of the CGM system [72].

Adequate treatment of pediatric T1DM has gradually shifted from HbA1c target values to time in range (TIR), which is defined based on CGM glycemia from 3.9 to 10 mmol/L [19]. Furthermore, good glycemic control is considered to be a TIR of more than 70%, a time below range (TBR) below 5%, and a time above range (TAR) below 30% with as little glycemic variability as possible (below 36%) [19]. In addition, certain age sub-groups of patients with T1DM can have a more aggressive approach to insulin therapy and, as a new CGM goal, spend more time in normoglycemia. A time in tight range (TITR) from 3.9 mmol/L to 7.8 mmol/L is increasingly set, which indicates the increasing reliability of the CGM system [75]. Previous studies indicated the importance of preventing hypoglycemia as one of the leading causes of acute complications, but recent studies in the pediatric population with T1DM indicate the importance of preventing hyperglycemia and glucovariability, which significantly affects cognitive and brain development [76,77]. All this extensive knowledge about the importance of CGM technology in the prevention of acute and chronic complications in T1DM should gradually be implemented in additional efforts to elucidate the complex cascade of MS and the occurrence of cardiovascular complications in children and adolescents with obesity.

## 5. Methods

This narrative review used a search of electronic databases of scientific publications from Medline/PubMed, Web of Science, and Scopus, and all studies were included until September 1, 2024. The search was based on the following keywords: “continuous glucose monitoring”, “obesity”, “children”, and “adolescents”. The inclusion factor was studies that were conducted on children and adolescents with obesity under the age of 18, and the language used was English. The search results generated 55 studies, and after a detailed review by the authors, 7 cross-sectional studies and 3 randomized intervention studies were included in the final version.

## 6. CGM in Children and Adolescents with Obesity

After an extensive review of the literature, we selected 7 cross-sectional studies that studied the role of the CGM system in children and adolescents with obesity (Table 2). According to the published data, the studies used relatively heterogeneous CGM systems from different manufacturers and models [78,79,80,81,82,83,84,85,86,87,88]. So far, in all of the studies, the use of CGM sensors is safe in the pediatric population with obesity, and no side effects have been published that differ from those described so far in other populations [78,86]. All the mentioned studies used the CGM sensor for a relatively short period of time (1 to 8 days), and the average age of the subjects was similar, with a relatively good distribution by sex. The majority of the studies were primarily based on the observation of glucose metabolism disorders and, depending on the study, described different incidences. Zou et al. showed a relatively high prevalence of glucose impairment (9 patients, 11.4%), of which 2 (2.5%) patients met T2DM criteria [78]. Similar findings were demonstrated on 17 subjects with obesity from the United States, which had nocturnal (from 4:00 to 6:00 AM) fasting glycemic values above 5.6 mmol/L, which was not seen during the OGTT [79]. Furthermore, in the study by Schiaffini et al. on 30 children with obesity, CGM glucose values showed a significant difference in specificity (92 vs. 52%) and sensitivity (95 vs. 90%) compared to the standard OGTT [81]. In addition to this, a Chinese study on 79 children and adolescents with obesity showed that 11 subjects (13.9%) had nocturnal episodes of hypoglycemia [78]. It is important to emphasize that the higher prevalence of nocturnal hypoglycemia can possibly be influenced by the compression artifact, which is a well-known cause of incorrect glucose readings on the CGM [89]. All of the above emphasizes the possible significant importance of using the CGM system in the assessment of glycemic disorders in children and adolescents with obesity, but additional studies with more subjects and a significantly longer follow-up period are needed in order to fully clarify all the benefits of CGM.

IR is the next very important link in clarifying the complex history of metabolic disorders in children and adolescents with obesity [2]. Multiple studies demonstrated a connection between CGM values and parameters of IR, including an Italian study that showed a significant correlation between HOMA-IR and basal insulin [79,81,82]. Further studies reported similar results of the association between glucovariability and IR but did not demonstrate an association between MS and individual components of MS (abdominal obesity, lipid profile, and blood pressure) [82]. The link between glucovariability and markers of low-grade inflammation (IL-6 and adiponectin) as an important factor in the development of MS was also not shown in the same cross-sectional study [82]. Given that so far only one study has investigated the impact of CGM values in the assessment of MS, it would be important to further clarify the impact of CGM in predicting the onset and progression of MS in future research.

In addition to this, so far, two research groups have conducted intervention randomized studies on the pediatric population with obesity for 12 weeks, during which CGM systems were also used (Table 3) [84,85,86]. In the first study, the influence of a high-protein breakfast on CGM values was examined, and it was shown that it statistically significantly reduced glycemic peaks (*p* < 0.03) and the time spent above the range (*p* < 0.03) [84]. The second study was more based on the influence of time-restricted eating to 8 h a day on the variability of glycemia in pediatric subjects; the main strength of the study was the use of the CGM system for a long period of time for the first time [85,86]. In a period of 13 weeks using the CGM system, 88% of subjects had a positive opinion about the use of the sensor, and it was concluded that the use of CGM combined with a nutritional report could be an adequate tool for monitoring glycemic excursion in children and adolescents with obesity [86]. Finally, a recent study demonstrated the interesting use of CGM in an intervention study with liraglutide in children and adolescent populations with obesity [88]. During the drug trial, there was a significant improvement in glucose metabolism, and the TITR significantly increased after the use of liraglutide for 3 months (91.76 vs. 94.18%, *p* = 0.048) [88]. This finding can be a useful tool and indicator of an adequate therapeutic response in the pediatric population with obesity.

## 7. Possible Future Perspective

CGM systems have their place in the diagnosis and monitoring of pediatric patients with obesity. Until now, studies conducted on the pediatric population have numerous limitations. First, most of the cross-sectional studies had an extremely small sample of patients with a short period of CGM system use. Additionally, most studies did not take into account puberty status, which can also affect physiological IR and glucovariability. The development of IR occurs at an early age and is associated with rapid growth in infants but also with the earlier onset of idiopathic precocious puberty [90]. These specific groups of children and adolescents have an increased risk of developing metabolic complications. It is extremely important in future research to qualitatively distinguish the impact of pubertal status and also the time and dynamics of the onset of puberty on glucose metabolism in children and adolescent populations with obesity using CGM systems [90]. Secondly, there was extreme heterogeneity in the inclusion criteria in the studies and the definition of obesity, which could also affect the reproducibility and comparability of results from published studies investigating glucose metabolism. Only recently has clear normative glucose sensor data for a healthy population been published, and this result can be a benchmark for future studies in populations with obesity [74]. Additionally, it would be interesting in future research to further clarify the effects of various unhealthy eating habits and a sedentary lifestyle on glycemic curves in children and adolescents with obesity in home conditions without previous interventions and education. Furthermore, it would be beneficial to clarify the influence of glycemic variability in subjects with obesity on the onset and progression of MS. Finally, we advise in the future, to conduct large-scale prospective studies in which CGM systems would be used in intervention studies and thus observe changes in glycemic values after various sports activities and dietary changes in order to find the most adequate and efficient way to treat extremely demanding pediatric obesity.

## 8. Conclusions

The CGM system has become crucial in today’s adequate treatment of diabetes, and the CGM technology has advanced significantly and is gradually entering other areas and diseases. Pediatric obesity is a leading public health problem in the world today and is increasingly the focus of today’s scientific community. The role of the CGM system has not yet been fully clarified in children and adolescents with obesity, and there is a need for additional research to further elucidate the early disorder of glucose metabolism. The CGM system can be an excellent tool in everyday clinical practice, not only in adequate diagnosis but also in various interventions, such as in the adjustment of nutritional habits and inadequate physical activity, which can be a guide in the treatment of pediatric obesity. At the end of this narrative review, we would like to point out that multicenter longitudinal randomized studies are needed that will additionally strengthen the importance of CGM systems in everyday clinical practices, which would help CGM to gradually be included in the recommendations of respectable pediatric societies.

## Figures and Tables

**Table 1 diagnostics-14-02801-t001:** Current CGM approved for children and adolescents.

CGM Type	Manufacturer	Approved for Age (yrs)	Size (mm)	Sensor Duration (Days)	Calibration (Yes/No)	Update of Glucose Value (min)	MARD Value for Children and Adolescents (%)
Dexcom G6	Dexcom	>2	45.7 × 30.5 × 15.2	10	no	5	7.7–10.1 [66,67]
Dexcom G7	Dexcom	>2	24 × 27.3 × 4.6	10	no	5	8.1–9 [58]
Eversense E3	Eversense	-	37.6 × 48 × 8.8	180	yes	implantable	9.1 [68]
FreeStyle Libre 2	Abbott	>4	5 × 35	14	no	1	9.7 [69]
FreeStyle Libre 3	Abbott	>4	2.9 × 21	14	no	1	9.4 [70]
Guardian 3	Medtronic	>2	19.3 × 11.4 × 9.7	7	yes	5	10.9 [71]
Guardian 4	Medtronic	>7	38 × 67 × 52	7	no	5	-
Medtrum S9	Medtrum	>2	29.8 × 17.8 × 5.2	14	no	2	-
Wellion sensor	Wellion	>4	35 × 21.8 × 6.8	10	no	5	-

CGM, continuous glucose monitoring; MARD, mean absolute relative differences.

**Table 2 diagnostics-14-02801-t002:** Cross-sectional clinical studies with CGM in children and adolescent populations with obesity.

Study	CGM Type	Duration CGM	Sample Size	Sex Female/Male	Mean Age (yrs)	Study Design	Main Outcomes
Zou et al. [78]	Medtronic MiniMed	24 h	79	25/54	10.53 ± 2.14	Assessment of glucose metabolism before, during, and after OGTT.	13.9% of subjects had nocturnal hypoglycemia. Meanwhile, 11.3% of the subjects met the criteria for glucose metabolism disorders, including T2DM.
Choudhary et al. [79]	Medtronic IPro	-	17	14/3	13.4 ± 2.7	Assessment of glucose excursions in children with prediabetes vs. IR.	Subjects with IR had glucose elevations during the night, which were not observed during the standard OGTT.
Chan et al. [80]	Medtronic IPro	72 h	98	63/35	-	Relationship between CGM values with HbA1c and OGTT.	CGM is equally valuable in predicting glucose metabolism disorders.
Schiaffini et al. [81]	Medtronic IPro2	48–72 h	30	15/15	12.87 ± 2.19	Relationship between CGM values and progression of NAFLD.	Degree of liver fibrosis significantly correlates with the CGM peak value.
Kaya et al. [82]	Medtronic Guardian	24 h	50	31/19	13.9 ± 2.3	Assessment of CGM values with and without MS.	There is no significant difference in the degree of glycemic variability in MS subjects in comparison with subjects without MS, as well as other components of MS.
Chylinska-Fratczak et al. [83]	Medtronic IPro	6 days	46	21/25	13.4 ± 2.5	Assessment of hyperglycemia.	Only 1% of subjects had higher CGM values than 7.7 mmol/L. CGM can possibly be a diagnostic tool for pre-diabetes.
Apperley et al. [87]	Dexcom G6	8 days	38	-	14.3	Assessment of CGM value vs. OGTT.	All subjects had a normal OGTT, but while wearing the CGM system, significantly less TIR was observed compared to healthy subjects.

CGM, continuous glucose monitoring; OGTT, oral glucose tolerance tests; IR, insulin resistance; HbA1c, glycated hemoglobin; NAFLD, non-alcoholic fatty liver disease; MS, metabolic syndrome; TIR, time in range.

**Table 3 diagnostics-14-02801-t003:** Randomized intervention clinical studies with CGM in children and adolescent populations with obesity.

Study	CGM Type	Duration CGM	Sample Size	Sex Female/Male	Mean Age (yrs)	Study Design	Main Outcomes
Bauer et al. [84]	Medtronic IPro	2 × 24 h	28	-	19 ± 1	Assessment of CGM values pre and post-study in high-protein vs. normal-protein breakfast.	High-protein breakfast improves glycemic control in comparison with a normal-protein breakfast.
Vidmar et al. [85] and Naguib et al. [86]	Dexcom G6	13 weeks	50	36/14	16.43 ± 1.17	Randomized control study to evaluate time-restricted eating and the role of CGM.	CGM can possibly be a key factor for tracing adherence to time-restricted eating. The study did not show significant differences in the variability of glycemia in the group with time-restricted enacting when compared to the control group.
Apperley et al. [88]	Dexcom G6 and FreeStyle Libre Pro	9.12 days	24	14/10	14.7	Effect of liraglutide on glycemic control.	Liraglutide significantly improved glucose levels in the normal range.

CGM, continuous glucose monitoring.

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
