# Peer review of "Continuous Glucose Monitoring—New Diagnostic Tool in Complex Pathophysiological Disorder of Glucose Metabolism in Children and Adolescents with Obesity"

_diagnostics, 2024, doi:10.3390/diagnostics14242801_

Round 1

Reviewer 1 Report

Comments and Suggestions for Authors

This review is a valuable contribution to the field. It provides a thorough overview of the current knowledge on CGM in pediatric obesity and identifies important areas for future research. With minor revisions to enhance critical analysis and contextual background, it will be an excellent resource for clinicians and researchers, however, only a few comments were found 

Inconsistent Definitions and Criteria:The review highlights different definitions and criteria for obesity and metabolic syndrome across studies. This inconsistency can affect the comparability of results. A more thorough discussion of the implications of these varying definitions would be useful. Also Lack of Consideration for Puberty,The review does not adequately address the impact of puberty on insulin resistance and glucose variability. Including a discussion on how puberty might affect the outcomes of CGM studies in children and adolescents would add depth to the analysis

Author Response

We are thankful for the comments and remarks that we received by your respectable reviewers. We hold that these comments provided a good blueprint for the more elaborated approach towards this exquisitely complex topic.

In this revision we gave our best effort to address every question and comment that your respectable reviewers had. A vast majority of suggestions were adopted and implemented throughout the text.

  1. We would like to address the following comment:
  • This review is a valuable contribution to the field. It provides a thorough overview of the current knowledge on CGM in pediatric obesity and identifies important areas for future research. With minor revisions to enhance critical analysis and contextual background, it will be an excellent resource for clinicians and researchers, however, only a few comments were found. Inconsistent Definitions and Criteria:The review highlights different definitions and criteria for obesity and metabolic syndrome across studies. This inconsistency can affect the comparability of results. A more thorough discussion of the implications of these varying definitions would be useful. Also Lack of Consideration for Puberty, The review does not adequately address the impact of puberty on insulin resistance and glucose variability. Including a discussion on how puberty might affect the outcomes of CGM studies in children and adolescents would add depth to the analysis”

Thank you very much for your valuable comment. As stated in the introduction of our manuscript, the definition of obesity is extremely heterogeneous. Additionally, there are approximately 40 different criteria are used to evaluate MS in children and adolescents. In order to successfully compare study results, it is necessary to establish stricter definitions of obesity and consequently MS. Furthermore, IR occurs at an early age and is associated with rapid growth in infants, but also with the earlier onset of idiopathic precocious puberty. This specific groups of children and adolescents have an increased risk of developing metabolic complications. Unfortunately, the stage of puberty has only been shown in a small number of studies, and its influence and specific subgroups have not been analyzed. We would like to emphasize the need for further research that will demonstrated glucose variability in different stages of puberty using CGM.

Addition to the dicussiom can be found on page 10.

Reviewer 2 Report

Comments and Suggestions for Authors

The topic of using continuous glucose monitoring (CGM) systems in children and adolescents with obesity is highly relevant as childhood obesity and its associated metabolic disorders are a significant global concern. The article addresses the timely issue of applying innovative technologies for diagnosing and managing these disorders, aligning with current trends in clinical practice. The article is well-structured, with a logical division into sections. The language is predominantly scientific; however, some parts are overly dense with terminology, which might make it harder for a broader medical audience to comprehend. The authors emphasize cutting-edge technologies, such as CGM, which are innovative for pediatric practice. The article incorporates recent research and recommendations, including ISPAD 2022, confirming the modernity of the methods employed. The discussion is based on the results of the selected studies. It is comprehensive but does not always critically analyze the limitations of the available data, such as small sample sizes or short study periods. The discussion generally aligns with the main text but tends to make generalizations that are not always supported by sufficiently robust evidence. The conclusions logically follow from the analyzed data; however, they lack emphasis on the need for long-term randomized studies to fully confirm the effectiveness of CGM. Overall, the article is a well-crafted review on a promising topic but would benefit from certain, albeit optional, clarifications and critical analysis to enhance its scientific value.

Author Response

We are thankful for the comments and remarks that we received by your respectable reviewers. We hold that these comments provided a good blueprint for the more elaborated approach towards this exquisitely complex topic.

In this revision we gave our best effort to address every question and comment that your respectable reviewers had. A vast majority of suggestions were adopted and implemented throughout the text.

We would like to address the following comment:

  • The topic of using continuous glucose monitoring (CGM) systems in children and adolescents with obesity is highly relevant as childhood obesity and its associated metabolic disorders are a significant global concern. The article addresses the timely issue of applying innovative technologies for diagnosing and managing these disorders, aligning with current trends in clinical practice. The article is well-structured, with a logical division into sections. The language is predominantly scientific; however, some parts are overly dense with terminology, which might make it harder for a broader medical audience to comprehend. The authors emphasize cutting-edge technologies, such as CGM, which are innovative for pediatric practice. The article incorporates recent research and recommendations, including ISPAD 2022, confirming the modernity of the methods employed. The discussion is based on the results of the selected studies. It is comprehensive but does not always critically analyze the limitations of the available data, such as small sample sizes or short study periods. The discussion generally aligns with the main text but tends to make generalizations that are not always supported by sufficiently robust evidence. The conclusions logically follow from the analyzed data; however, they lack emphasis on the need for long-term randomized studies to fully confirm the effectiveness of CGM. Overall, the article is a well-crafted review on a promising topic but would benefit from certain, albeit optional, clarifications and critical analysis to enhance its scientific value.

Thank you very much for your valuable comment; we further emphasize the limitations of the studies and the use of CGM systems in the obese pediatric population. We fully agree with the reviewer that additional, better designed (long-term randomized studies) studies with adequate samples are needed to fully clarify the benefits of CGM systems in obese children and adolescents.

Addition to the discussion can be found on page 4 and 10.